# Sustainable purification-free synthesis of *N*–H ketimines by solid acid catalysis

**Shintaro Shibata** [1] ✉ **& Makoto Onaka** [2] ✉

Among imines, nitrogen-unprotected ketimines (*N*–H ketimines) are valuable precursors to nitrogen-containing compounds. However, their applications are limited compared with those of nitrogen-protected ketimines (*N*–R ketimines) owing to difficulties in synthesis and purification. We develop remarkably simple methods for synthesizing and isolating high-purity *N*–H ketimines and *N*–H ketimine hydrochlorides via the dehydration–condensation of ketones with stoichiometric ammonia, generated in situ with inorganic solid acid catalysts. Our methods offer exceptionally broad substrate scope, use non-toxic and reusable catalysts, do not require tedious synthetic steps, are inexpensive, and are suited to large-scale production. Additionally, ketones can be easily converted to *N*–R ketimines, α-aminonitriles, and hydantoins in one pot via *N*–H ketimines. All products can be isolated by filtration or concentration; other purification methods, such as column chromatography, are not required. This study guides the advancement of the design of new transition metal catalysts and pharmaceutical synthesis.

Imines are valuable intermediates in various fields, including the chemical and pharmaceutical industries[1–3]. Nitrogen-unprotected imines (*N*–H imines) are particularly versatile compared with nitrogen-protected imines (*N*–R imines), because they can be converted to a variety of useful compounds, in particular, primary amines, which are crucial building blocks in organic synthesis[4]. Among the different *N*–H imines, *N*–H aldimines (RCH = NH) derived from aldehydes are difficult to isolate due to their thermal instability and facile self-polymerization[5,6]. By contrast, *N*–H ketimines (R$_2$C = NH) obtained from ketones are more stable than *N*–H aldimines and can be isolated, making them promising precursors for primary and secondary amines, *N*–R ketimines, and supporting ligands for metal complex catalysts (Fig. 1a). However, an efficient method for synthesizing various high-purity *N*–H ketimines has yet to be discovered.

Ketones, nitriles, azides, amines, and aminonitriles are used as starting materials for producing *N*–H ketimines (Fig. 1a, first to fourth quadrants; see refs. 4, 7. for details). Although the denitrogenation of azides[8,9], oxidation of amines[10], pyrolysis of aminonitriles[11], and syntheses from ketones—including classical aza-Wittig reactions[12,13],

TMS-imine formation using MN(TMS)$_2$ (M = Li, K)[14,15], and multistep reactions via oximes[16] or nitroimines[17]—can afford a few types of *N*–H ketimines, these methods are not versatile and often involve complex operations. Common methods for adding organometallic reagents to nitriles involve strongly basic and water-sensitive conditions[18–20]. The most direct method for synthesizing *N*–H ketimines is the dehydration–condensation of ketones with NH$_3$. This reaction is performed under the control of thermodynamic equilibria (see Section 7 in Supplementary Information for calculation of ΔG), where the reactant side is generally favored over that of the products unless the products are stabilized by intramolecular hydrogen bonds[21–24]. Therefore, the dehydration–condensation of ketones with NH$_3$ is limited to classical synthesis under high-temperature and high-pressure conditions (120 °C, > 50 bar)[25] or using strongly acidic dehydrating agents such as TiCl$_4$[26,27]. These methods require purification using distillation, extraction with organic solvents/water, or column chromatography to isolate the *N*–H ketimines, rendering them unsuitable for thermally unstable and hydrolytically sensitive *N*–H ketimines; only substrates, such as the relatively stable *N*–H diaryl ketimines, can be

[1]Research Foundation ITSUU Laboratory, Kanagawa, Japan. [2]Faculty of Life Sciences, Tokyo University of Agriculture, Tokyo, Japan. ✉e-mail: sshibata@itsuu.or.jp; conaka@g.ecc.u-tokyo.ac.jp

isolated (Fig. 1a, third and fourth quadrants; see Section 3.8 in Supplementary Information for hydrolysis test of $N$–H diaryl ketimine during column chromatography). Sc(OTf)$_3$ and tetrabutylammonium fluoride (TBAF) have recently been reported as catalysts for synthesizing $N$–H ketimines[28,29]. Although these systems allow the isolation of hydrolysis-resistant $N$–H ketimines via rapid purification using silica gel-flash column chromatography, their substrate scope remains limited. In our previous studies, we reported the synthesis of $N$–H ketimines from ketones and excess ammonia using inorganic solid acids, in which the dehydration–condensation equilibrium was shifted toward the product side[7,30]. However, owing to the thermodynamic instability of $N$–H ketimines, it was not possible to quantitatively synthesize a wide range of substrates.

In this work, we show that inorganic solid acids enable the dehydration–condensation of ketones with stoichiometric ammonia, generated in situ from hexamethyldisilazane (HMDS), to produce high-purity $N$–H ketimines under mild conditions. This method offers a broad substrate scope, uses inexpensive and reusable catalysts, and provides scalable access to $N$–H ketimines and their derivatives, such as $N$–H ketimine hydrochlorides, $N$–R ketimines, α-aminonitriles, and hydantoins.

## Results and discussion
### Development of key catalytic strategies
In this study, we developed a catalytic reaction system (Fig. 1b). First, NH$_3$ is spontaneously generated from H$_2$O and hexamethyldisilazane ((Me$_3$Si)$_2$NH, HMDS). Subsequently, the NH$_3$ condenses with the ketone activated on the solid acid catalyst to produce the corresponding $N$–H ketimine and H$_2$O. Byproduct H$_2$O immediately reacts with HMDS to produce stable hexamethylsiloxane ((Me$_3$Si)$_2$O) and a new NH$_3$ molecule. This reaction leads to an irreversible catalytic cycle. At the end of the reaction, the solid acid catalyst and low-boiling-point components, such as HMDS, (Me$_3$Si)$_2$O, and NH$_3$, are removed by simple filtration and concentration techniques from the reaction mixture to produce the desired high-purity $N$–H ketimine (I) (Fig. 1b, c). Surprisingly, $N$–H ketimines can also be easily converted to hydrochlorides (II) without producing NH$_4$Cl as a byproduct, through the methanolysis of residual HMDS, followed by N$_2$ gas-bubbling. Furthermore, this system enables the one-pot derivatization of $N$–H ketimines to $N$–R ketimines (III), α-aminonitriles (IV), and hydantoins (V) without conventional purification procedures (Fig. 1c).

### Screening of reaction conditions
We first compared the catalytic activities of various heterogeneous catalysts, including a range of inexpensive inorganic solid acids, for the syntheses of $N$–H ketimines from benzophenone 1a (Fig. 2a). The reaction of 1a with HMDS was conducted in a screw vial (2 mL) at 40 °C for 24 h under solvent-free conditions. Zeolites containing H$^+$ ions exhibit higher catalytic activity than those containing metal ions (e.g., Na-Y vs. H-Y). Among zeolites with comparable H$^+$ content (i.e., similar Si/Al ratios), such as H-Y (Si/Al = 14.5), H-Beta (Si/Al = 12.5), and H-Mor (Si/Al = 9), differences in acid strength arising from structural variations (in order of acid strength: H-Y < H-Beta < H-Mor)[31] indicate that zeolites with lower acid strength tend to afford higher yields. Moreover, for zeolites with the same framework structure, higher H$^+$ content (i.e., a lower Si/Al ratio) generally correlates with their increased activity. In contrast, no clear correlation between yields and specific surface area was observed for these zeolites. Although $N$–H ketimine 2a was not obtained using silica, regardless of its acidity or structure, a certain kind of the various alumina catalysts was able to quantitatively produce 2a. Silica-alumina (SiO$_2$-Al$_2$O$_3$) and ion-exchange montmorillonite clays (M-Mont; except for M = Na) afforded 2a in moderate-to-high yields. By contrast, the reaction barely proceeded in the presence of montmorillonite K10 (Mont K10) or the ion-exchange resins

(polymer-$p$-C$_6$H$_4$SO$_3$H and polymer-C$_6$H$_4$NMe$_2$). Among common salt reagents, Al$_2$(SO$_4$)$_3$ displayed the highest activity (yield > 99%).

Next, we shortened the reaction time to 1 h to compare the catalysis of aluminum-containing materials between γ-type aluminas, mesoporous aluminas, and Al$_2$(SO$_4$)$_3$. It turned out that sulfate-ion-modified mesoporous alumina ($meso$-Al$_2$O$_3$/SO$_4^{2-}$) was the most active catalyst. This $meso$-Al$_2$O$_3$/SO$_4^{2-}$ catalyst[32,33] containing mesopores was prepared from Al(O-$sec$-Bu)$_3$ and Al$_2$(SO$_4$)$_3$ using a sol-gel reaction based on a mesoporous alumina preparation method[34]. The acidic character of mesoporous alumina is enhanced by sulfate ions incorporated in the Al$_2$O$_3$ framework, and $meso$-Al$_2$O$_3$/SO$_4^{2-}$ was believed to work as an acid catalyst to activate ketones in the present reaction[32,33].

Although the reaction proceeds efficiently without additional solvents, the role of solvents should be investigated. Therefore, solvent screening was conducted using the best-performing $meso$-Al$_2$O$_3$/SO$_4^{2-}$ catalyst (Fig. 2b). The reaction under solvent-free conditions yielded a quantitative amount of $N$–H ketimine 2a, and similar yields were observed with low-polarity solvents, such as cyclohexane, benzene, and diisopropyl ether. By contrast, using solvents such as toluene, fluorobenzene, tetrahydrofuran (THF), 1,4-dioxane, and ethyl acetate led to a mild or moderate decrease in product yield. Chlorinated solvents, acetonitrile, and alcohols afforded significantly lower yields, while in highly polar aprotic solvents such as $N$,$N$-dimethylformamide (DMF), dimethyl sulfoxide (DMSO), nitromethane, and sulfolane, the reaction barely proceeded. In addition to solvent effects, increasing the reaction temperature from 40 to 60 or 80 °C enhanced the product yield. These findings indicate that solvent-free conditions are optimal for synthesizing 2a, and that low-to-medium polarity solvents (such as cyclohexane, benzene, and diisopropyl ether) can be a secondary option when needed. Additionally, a combination of highly polar ether-type solvents (THF and 1,4-dioxane) and slightly elevated reaction temperatures can be used as a third option.

### Mechanistic investigation
We next investigated the effects of adding organic bases on the formation of $N$–H ketimines to characterize the catalytic active sites (Fig. 2c). First, 20 mol% of various organic bases in benzene-$d_6$ was added to an NMR tube containing the reagents and the $meso$-Al$_2$O$_3$/SO$_4^{2-}$ catalyst necessary for the synthesis of $N$–H ketimines. The NMR tube was flame-sealed, and the resulting mixture was heated at 60 °C. Changes in the $N$–H ketimine yield were measured over time by $^1$H NMR analysis. In the absence of an organic base (red circles), the reaction in benzene-$d_6$ required 140 min to reach completion with the disappearance of ketone 1a. By contrast, in the presence of pyridine, 4-methylpyridine, 2,6-lutidine, 2,6-di($t$-butyl)pyridine, and proton sponge, the reaction appeared to reach saturation within 90–110 min. These results can be ascribed to the higher reaction rates, which are caused by a slight enhancement in the nucleophilicity of ammonia due to the added base. By contrast, the reaction rate decreased markedly when either Et$_3$N, 1,4-diazabicyclo[2.2.2]octane (DABCO), or 4-methoxypyridine was added, probably owing to the weakening of acid catalysis by the added strongly nucleophilic organic base. Interestingly, the addition of 4-dimethylaminopyridine (DMAP) completely suppressed the reaction. In contrast to DMAP, adding proton sponge as a proton scavenger had no effect on the rate, indicating that the catalytic active sites on the alumina surface are Lewis acid sites rather than the Brønsted ones.

A comparison of $meso$-Al$_2$O$_3$/SO$_4^{2-}$ (red circles) with acidic γ-Al$_2$O$_3$ (yellow squares) revealed that the activity of $meso$-Al$_2$O$_3$/SO$_4^{2-}$ is enhanced by the sulfate ions incorporated into the framework. This finding is consistent with the results of previous findings[32], which suggested that the incorporation of sulfate enhances catalytic activity. The $meso$-Al$_2$O$_3$/SO$_4^{2-}$ used in this study contains sulfur uniformly dispersed throughout the material, and is not simply a physical mixture of alumina and Al$_2$(SO$_4$)$_3$, as evidenced by IR-ATR, XRD, XPS, and

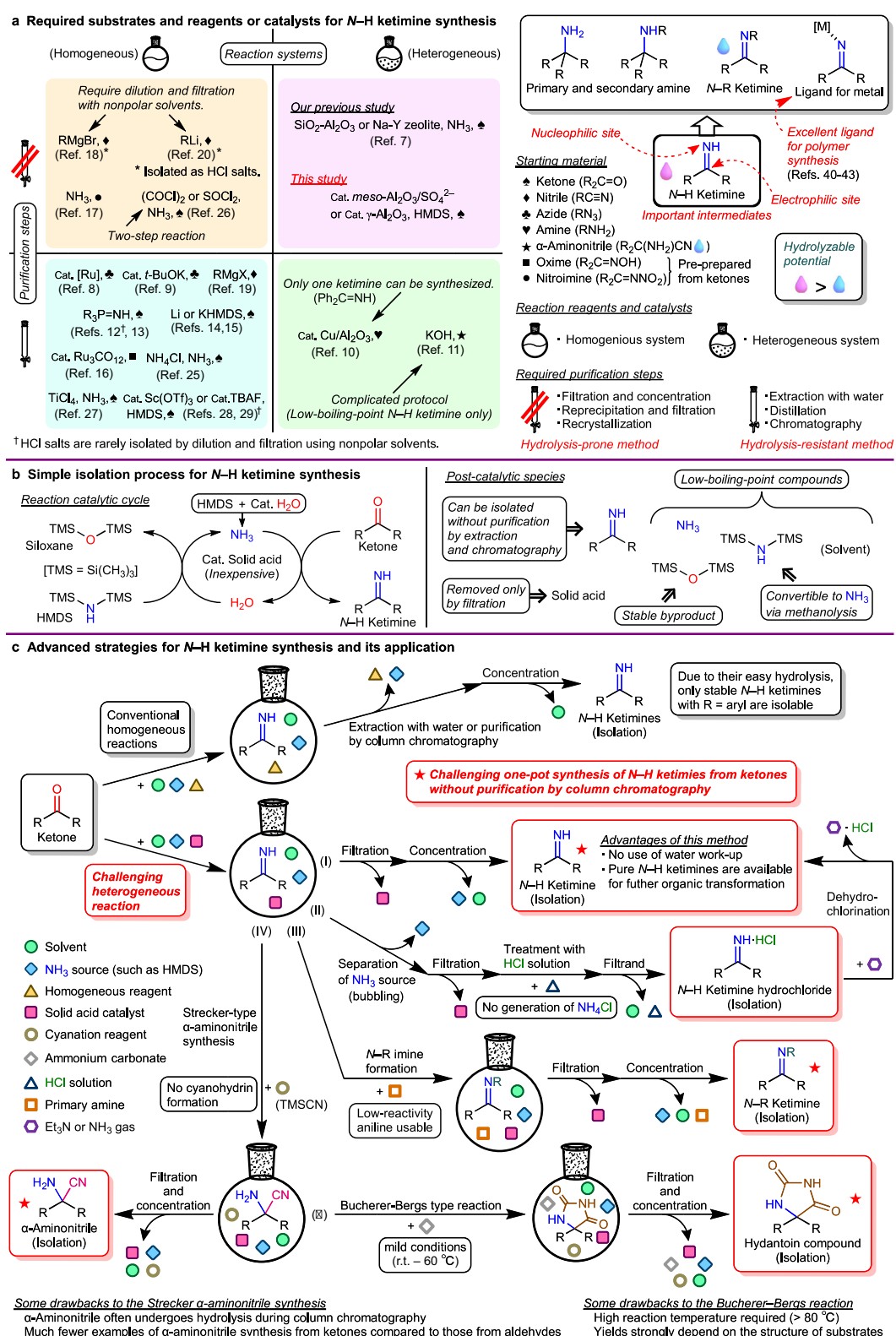

**Fig. 1 | Outline of the study. a** Features of *N*–H ketimine and key points from previous studies. **b** Simple isolation process for *N*–H ketimine synthesis on solid acids. **c** Advanced synthesis strategies using *N*–H ketimines. Abbreviations:

Cat. = catalyst; HMDS = ((CH$_3$)$_3$Si)$_2$NH; r.t. = room temperature; TMS = (CH$_3$)$_3$Si–; TMSCN = (CH$_3$)$_3$SiCN.

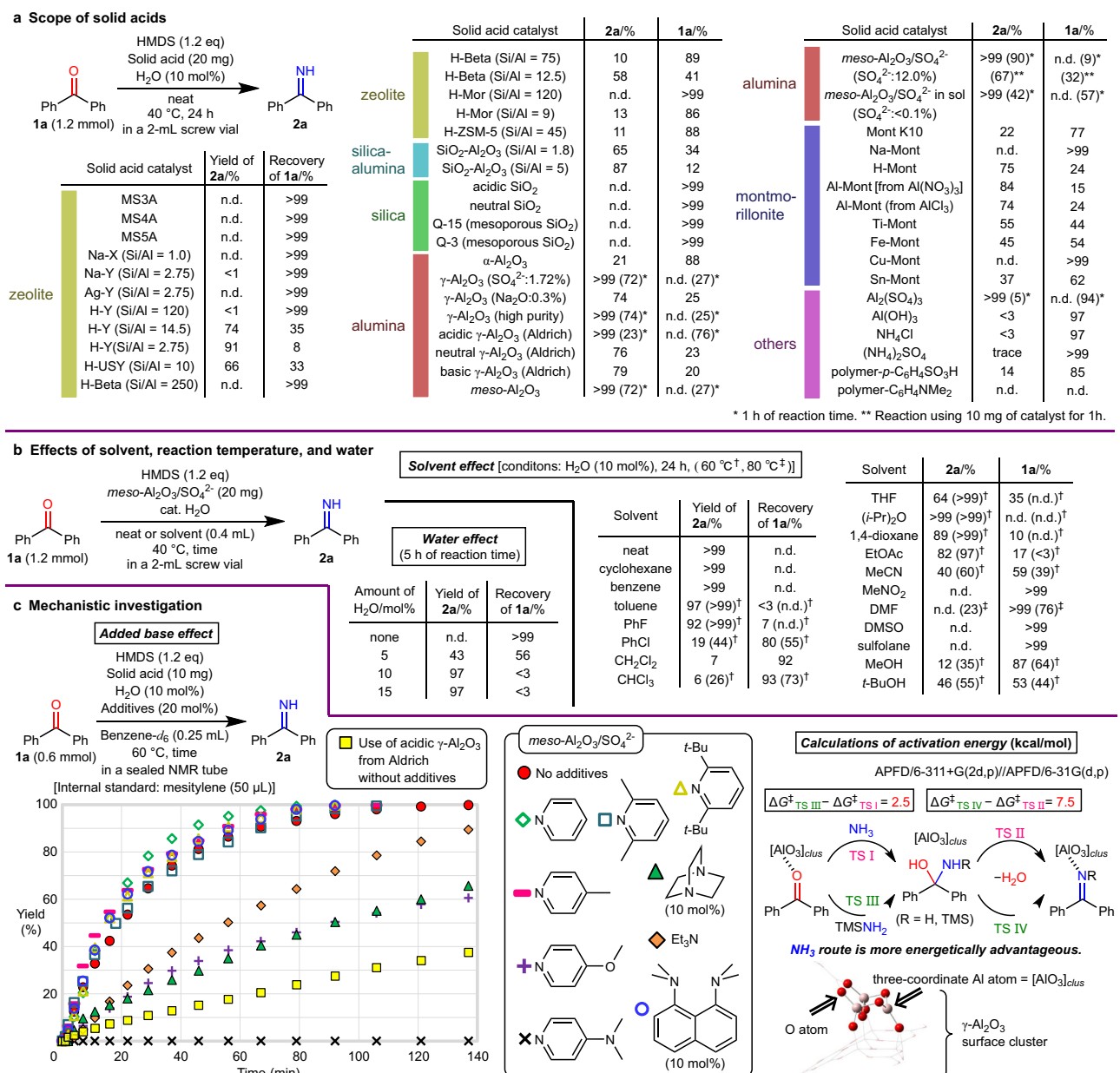

**Fig. 2 | Optimization of reaction conditions and mechanistic evaluation.**
**a** Catalyst screening to identify the most effective catalyst. **b** Effects of solvent, reaction temperature, and water. **c** Influence of bases and mechanistic insights through quantum chemical calculations. Abbreviations: THF = tetrahydrofuran; DMF = *N,N*-dimethylformamide; DMSO = dimethyl sulfoxide.

SEM analyses (see Sections 2.2 and 2.3 in Supplementary Information). Conventionally, the active Lewis acid sites on $\gamma$-Al$_2$O$_3$ have been believed to be three-coordinated Al atoms[35–37]. Therefore, the active sites of *meso*-Al$_2$O$_3$/SO$_4$$^{2-}$ with the highest activity are three-coordinated Al atoms[32]. Thus, we constructed a surface cluster model for $\gamma$-Al$_2$O$_3$ and calculated the activation energy for the nucleophilic addition of NH$_3$ and trimethylsilylamine (TMSNH$_2$), which are believed to originate from HMDS, to ketone **1a**. The addition process of NH$_3$ is accompanied by a lower-energy pathway than that of TMSNH$_2$ (Fig. 2c, see Section 8 in Supplementary Information for calculation details).

Notably, the reactions did not proceed in the absence of a small amount of H$_2$O (Fig. 2b). This result clearly indicates that HMDS does not directly attack the ketone. Instead, NH$_3$ generated from H$_2$O and HMDS reacts with the ketone to produce the *N*–H ketimine.

## Substrate scope for *N*–H ketimine synthesis

The scope of applicable ketones was investigated (Fig. 3a, b). In Fig. 3, the yields for the *N*–H ketimines synthesized unprecedentedly are marked with gold diamonds; those not synthesized from ketones are marked with silver squares; and those not synthesized directly from ketones in a single step are marked with copper circles. The reactivity depends greatly on the molecular structure of the ketone; therefore, the $\alpha$–$\delta$ reaction conditions were used for each substrate (conditions $\alpha$, $\beta$, and $\gamma$ for *N*–H aryl ketimines; conditions $\delta$ for *N*–H alkyl ketimines). For unstable *N*–H ketimines, which contained a trace of inseparable byproducts or were severely susceptible to hydrolysis even when only a trace of moisture was present, the $^1$H NMR yields were determined using 1,4-dioxane or mesitylene as internal standards. After the removal of the NH$_3$ generated via methanolysis of HMDS, the unstable *N*–H ketimines were treated with 1 M HCl/Et$_2$O to afford more

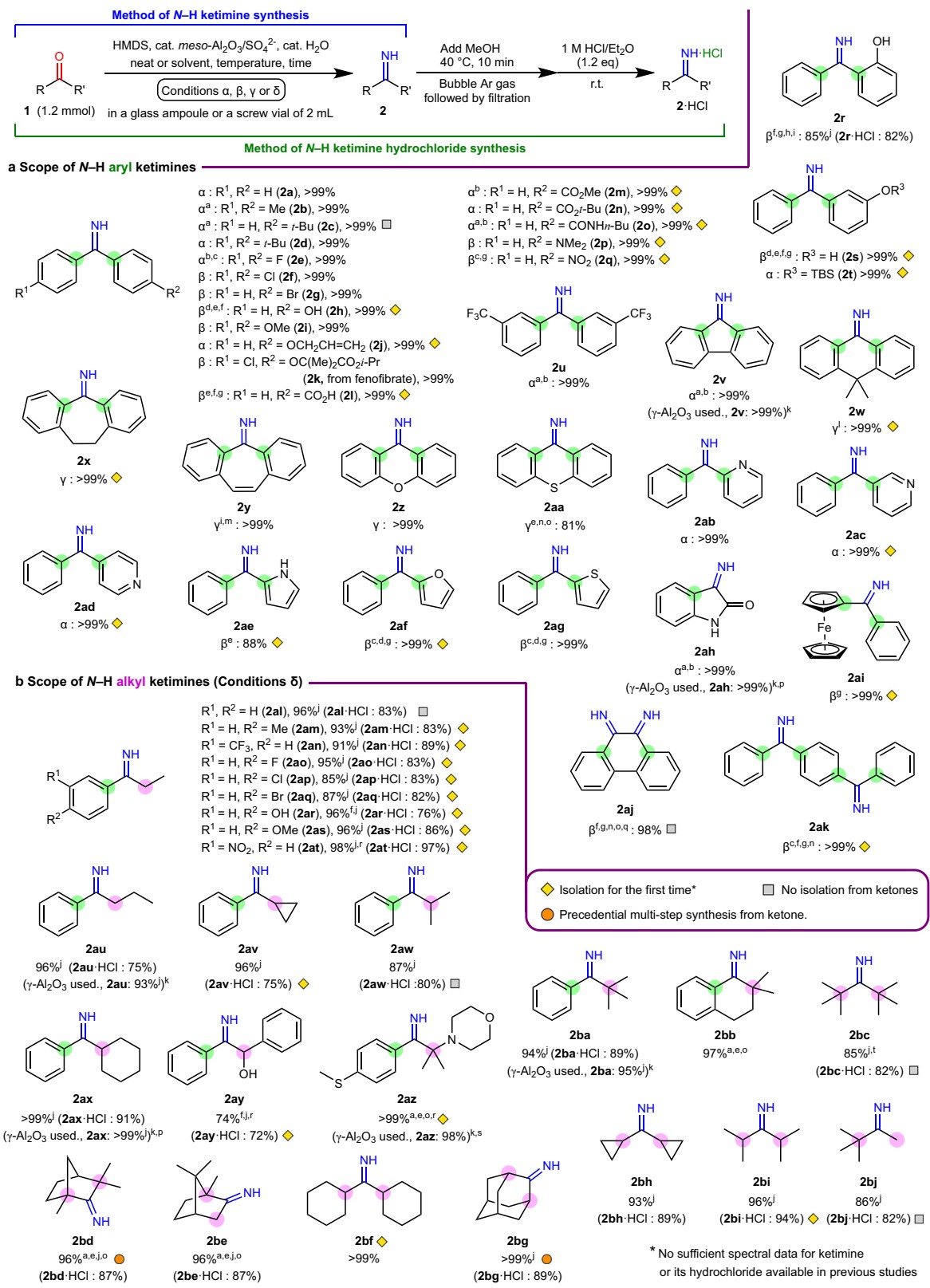

**Fig. 3 | Substrate scope for N–H ketimine formation.** All isolated yields are shown. **a** N–H aryl ketimines. **b** N–H alkyl ketimines. Conditions α: HMDS (1.2 eq), *meso*-Al₂O₃/SO₄²⁻ (20 mg), H₂O (10 mol%), neat, 60 °C, 24 h. Conditions β: HMDS (1.5 eq), *meso*-Al₂O₃/SO₄²⁻ (40 mg), H₂O (20 mol%), toluene (1.2 mL), 90 °C, 24 h. Conditions γ: HMDS (2.0 eq), *meso*-Al₂O₃/SO₄²⁻ (40 mg), H₂O (40 mol%), mesitylene (1.2 mL), 160 °C, 24 h. Conditions δ: HMDS (1.2 eq), *meso*-Al₂O₃/SO₄²⁻ (20 mg), H₂O (10 mol%), neat, 40 °C, 24 h. [a] HMDS (1.5 eq) and H₂O (20 mol%) were used. [b] Toluene (1.2 mL) was added. [c] Reaction temperature was 80 °C. [d] Solvent-free conditions. [e] Reaction temperature was 100 °C. [f] HMDS (2.4 eq) was used. [g] Catalyst (20 mg) was used.

[h] Mesitylene (1.2 mL) was used instead of toluene. [i] Reaction temperature was 140 °C. [j] NMR yield. [k] Acidic γ-Al₂O₃ (Aldrich) was used instead of *meso*-Al₂O₃/SO₄²⁻. [l] Reaction conditions: amyl ether (1.2 mL) instead of toluene, 180 °C, 48 h. [m] Dibutyl ether (1.2 mL) was used instead of mesitylene. [n] 1,4-Dioxane (1.2 mL) was used instead of toluene or mesitylene. [o] Reaction time was 48 h. [p] Catalyst (40 mg) was used. [q] Reaction temperature was 40 °C. [r] Toluene (0.2 mL) was added. [s] Conditions γ were applied at a modified temperature of 140 °C. [t] Reaction conditions: HMDS (2,8 eq), catalyst (100 mg), H₂O (100 mol%), and 60 °C.

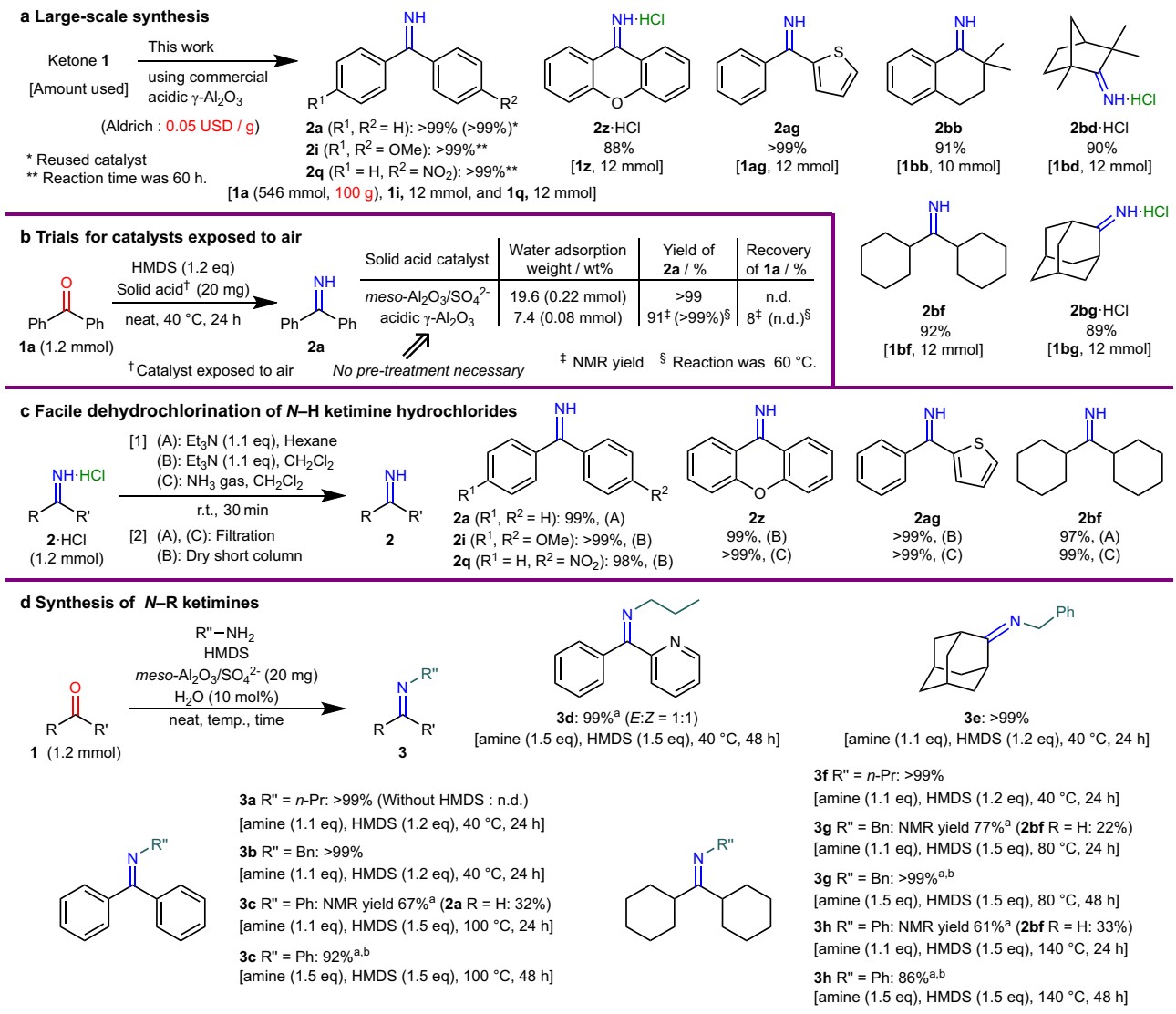

**Fig. 4 | Practical evaluation and synthetic applications of catalysts.** All isolated yields are shown. **a** Alternative use of commercial catalysts for large-scale synthesis. **b** Air exposure test for catalyst usability. **c** Facile dehydrochlorination of N–H ketimine hydrochlorides via salt metathesis. **d** Scope of synthesis of N–R ketimines.

[a] A 15 mL pressure-resistant glass tube was used. [b] Amine (1.1 eq) was first added and allowed to react for 24 h. Then, amine (0.4 eq) was added again and reacted further for 24 h.

stable N–H ketimine hydrochloride salts, which were easily isolated by crystallization (see Section 3.3 and 3.4 in Supplementary Information for details).

Various N–H diaryl ketimines were synthesized from the benzophenone derivatives (Fig. 3a). The N–H ketimines containing a wide variety of substituents, such as alkyls, halogens, hydroxyl, alkyl ether, silyl ether, carboxyl, ester, amide, tertiary amine, nitro, and CF₃ groups, were quantitatively produced (**2b–2u**). The N–H ketimines with pyridine, pyrrole, furan, and thiophene rings were also obtained in good yields. Notably, ketones **1ae** and **1af** were easily converted to N–H ketimines, although pyrrole[38] and furan[39] rings are generally unstable under acidic conditions. Furthermore, isatin **1ah**, which has various biological activities, and N–H ketimines with a ferrocene moiety were quantitatively obtained (**2ah**, **2ai**). The protocol allowed the efficient and facile production of bifunctional N–H ketimines from diketones **1aj** and **1ak**.

Next, we synthesized N–H alkyl ketimines, which are more difficult to synthesize than N–H aryl ketimines (Fig. 3b). The corresponding N–H alkyl aryl ketimines were successfully

obtained in high yields from the alkyl aryl ketones (**2al–2bb**). In particular, the successful formation of bulky N–H ketimines **2az–2bb** should be noted. Various N–H dialkyl ketimines **2bc–2bj** were also obtained. For example, we used one-step reactions to synthesize **2bd** and **2bg**, which were previously obtained via multistep reactions[15,17]. Another advantage of this protocol is that low-boiling-point N–H ketimines, such as **2bh**, **2bi**, and **2bj**, were easily isolated in high yields as their hydrochloride salts.

Recently, N–H dialkyl ketimines such as **2bc**, in which the imine part is surrounded by sterically hindered tertiary butyl substituents, have been recognized as the optimum ligands for transition metal catalysts, which can greatly enhance the copolymerization activities[40–43]. The conventional preparation of **2bc** requires utilizing hazardous t-BuLi, which is a very strong and highly pyrophoric base. The protocol described herein enables the safe, facile, and large-scale synthesis of **2bc** as well as that of various bulky N–H ketimines, which are expected to act as nitrogen-containing supporting ligands for various transition-metal-catalyzed reactions.

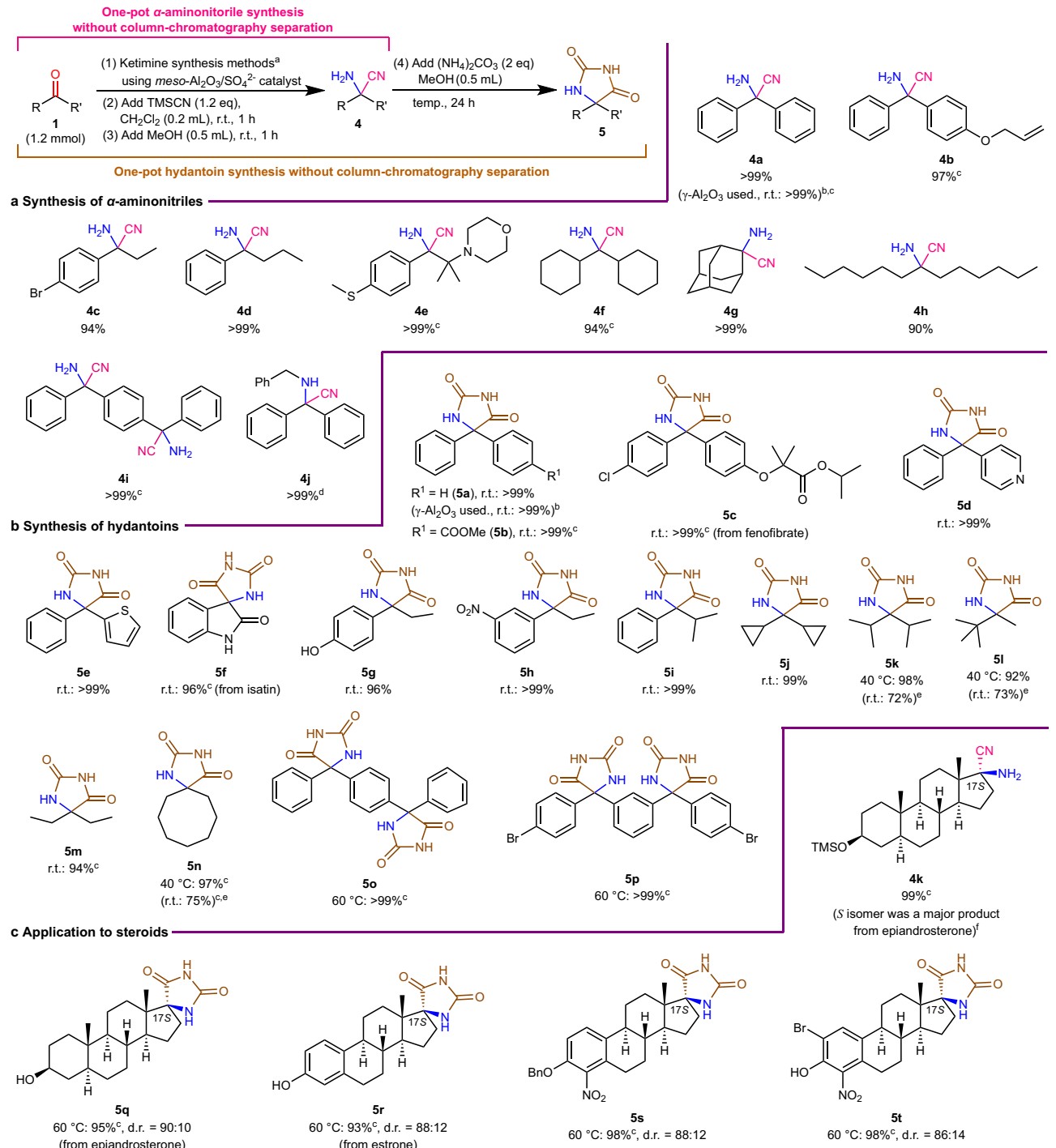

**Fig. 5 | Extended one-pot synthesis applications under mild conditions.** All isolated yields are shown. **a** Synthesis of α-aminonitriles. **b** Synthesis of hydantoin derivatives. **c** Application to steroids. [a] Under the reaction conditions based on each N−H ketimine synthesis shown in Fig. 3. [b] Acidic γ-Al₂O₃ (Aldrich) was used instead of meso-Al₂O₃/SO₄²⁻. [c] Conditions such as reagent equivalents and amounts/types of solvents changed from the standard conditions; see the Supplementary Information for details. [d] Containing 5% of **4a** as a byproduct. [e] NMR yield. [f] The diastereomeric ratio of hydantoin **5q** determines that the main isomer of aminonitrile **4k** has the S configuration at the C-17 site.

## Expansion of synthetic utility

To develop a more practical method, we performed a few large-scale experiments using commercially available acidic γ-Al₂O₃ (0.05 USD/g) (Fig. 4a). Based on the results shown in Fig. 3a, b, we selected ketones, including those with low reactivity. The reaction conditions shown in Fig. 3b were adopted, albeit on a ten-times larger scale. All target products were obtained in high yields of 88–99%. After the standard work-up procedure, N−H ketimine **2a** was quantitatively produced from 100 g of ketone **1a** without any decrease in the yield. After the recovery and subsequent calcination of the catalyst from the first experiment, a second run also produced a quantitative yield of **2a**. This result shows that the catalyst can be reused even in such large-scale syntheses. The results also confirmed that the γ-Al₂O₃ and meso-Al₂O₃/SO₄²⁻ catalysts, exposed to humid air for more than three months,

quantitatively promoted the formation of *N*–H ketimines at 60 and 40 °C, respectively (Fig. 4b). In other words, if the catalyst surface contains sufficient moisture, catalyst pre-activation by heating in a vacuum or $H_2O$ addition is not required. These results suggest that commercially available and inexpensive γ-$Al_2O_3$ is a promising option. By contrast, *meso*-$Al_2O_3$/$SO_4^{2-}$ is the optimal catalyst for synthesizing *N*–H ketimines at lower temperatures and within a reduced time in relation to acidic γ-$Al_2O_3$ (Fig. 2a, c).

Following the efficient synthesis of *N*–H ketimines, conversion to their hydrochloride salts provides a practical means for their long-term storage. These salts are stable under anhydrous conditions and free *N*–H ketimines can be regenerated in nearly quantitative yield upon treating the hydrochloride salts with $Et_3N$ or $NH_3$ gas at 20–25 °C for 30 min, followed by solvent rinsing/concentration or passing through a dried short column (Fig. 4c). These findings demonstrate that this method enables scalable synthesis with simple operation, and that the method is adaptable to diverse reaction conditions for synthetic applications.

### Versatile applications via one-pot synthesis
We successfully achieved the one-pot synthesis of *N*−R ketimines, α-aminonitriles, and hydantoin compounds from ketones via intermediate *N*–H ketimines (Figs. 4d, 5a, b). To generate *N*–R ketimines, various amines were added to the standard reaction mixtures for the formation of *N*–H ketimines. For example, when propylamine or benzylamine was used along with ketone **1a**, the reaction proceeded well to quantitatively produce **3a** or **3b** at a low temperature of 40 °C. Even when a combination comprising less-nucleophilic aniline was used with bulky dicyclohexyl ketone **1bf**, the desired *N*–R ketimine **3h** was isolated in a high yield (86%) under appropriate reaction conditions. Interestingly, in the absence of the HMDS, no condensation of ketone **1a** with pro-pylamine took place to afford the *N*–Pr ketimine. Furthermore, when **3c**, **3g**, and **3h** were synthesized within a shorter reaction time (24 h), the corresponding *N*–H ketimine intermediates **2a**, **2bf**, and **2bf** were produced in 32%, 22%, and 33% yields, respectively. These intermediates were further treated with an additional 0.4 equivalents of the corresponding amine for another 24 h to afford **3c**, **3g**, and **3h** in 92%, >99%, and 86% yields, respectively. This result clearly indicates that *N*–R keti-mines are formed from ketones via the essential *N*–H ketimine intermediate.

Different types of high-purity α-aminonitriles were synthe-sized in almost quantitative yields by simply adding trimethylsilyl cyanide (TMSCN) to the reaction mixture at 20–25 °C without performing any complicated separation operations after the for-mation of *N*–H and *N*–R ketimines (**4a**–**4k**). In particular, focus should be placed on the successful preparation of aliphatic α-aminonitriles such as **4h** that are susceptible to hydrolysis on silica gel columns (see Section 5.2 in Supplementary Information). Generally, α-aminonitriles from ketones are more difficult to synthesize than those obtained from more electrophilic aldehydes[44]. Nonetheless, our method can be used for the facile synthesis of various α-aminonitriles from any ketone having a linear alkyl group, branched alkyl, or an aromatic ring on either or both sides of the carbonyl functionality without cumbersome purification procedures. Moreover, α-secondary aminonitriles **4j** can be prepared via a one-pot reaction involving the direct addition of TMSCN to *N*–R ketimines obtained from ketones and primary amines.

The one-pot conversion of ketones to different hydantoin com-pounds (**5a**–**5t**) was efficiently achieved under very mild conditions (nearly ambient pressure and temperature; see Section 6.1 of Supple-mentary Information for optimization of reaction conditions). Nota-bly, the syntheses of **5f** and **5q**–**5t** linked to a steroidal skeleton have

not been previously reported; those compounds are expected to exhibit pharmacological properties. The classical Bucherer−Bergs reaction has been used for the preparation of hydantoin compounds[45]. However, NaCN or KCN is required, and is more toxic than TMSCN at high temperatures (> 80 °C) in water/alcohol solvents. In addition, isolating the various hydantoins in excellent yields is difficult, as the yields vary significantly depending on the structure and functional groups of the substrates (see Section 6.3 in Supplementary Informa-tion for a comparison of conventional methods). This is because (i) the *N*–H ketimine and α-aminonitrile intermediate derived from the ketone are unstable and easily hydrolyzed, and (ii) the highly polar hydantoin compounds are separated from the alkali metal salt byproducts with difficulty. By contrast, the only successful synthesis involving TMSCN was obtained using liquid $NH_3$ and $CO_2$ gas under pressure in the presence of the $Ga(OTf)_3$ catalyst at − 78 °C under severely anhydrous conditions[46]. Compared with these methods, in our protocol, various hydantoins (including **5o**–**5t**, which have very low solubility in many common organic solvents) can be synthesized under mild conditions and isolated through simple filtration and concentration procedures.

This study demonstrates that *N*–H ketimines, which are unstable and difficult to prepare, can be safely and easily synthesized using various ketones, an ammonia equivalent of HMDS, and a thermally and chemically stable inorganic solid acid. The *N*–H ketimines can be uti-lized as reaction intermediates for the synthesis of various valuable organic compounds, and as a unique framework for nitrogen-containing ligands that have not yet been examined. The results obtained in this study underscore the potential of transition-metal-catalyzed organic reactions and pharmacological studies.

## Methods
All experimental procedures are provided in the Supplementary Information.

## Data availability
All experimental procedures and compound characterization data (NMR, IR, and MS spectra) are provided in Supplementary Information. The Supplementary Information also contains the computational methods, detailed coordinate data from quantum chemical calcula-tions, and the analyses and discussions of the catalytic studies. Data supporting the findings of this manuscript are also available from the corresponding author upon request. Source data are provided in this paper.

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

## Acknowledgements

We thank Dr. Mitsuaki Ohtani (ITSUU Laboratory), Dr. Noritaka Chida (ITSUU Laboratory), Professors Kiyosei Takasu (Kyoto University), Tomohiko Ohwada (The University of Tokyo), Masayuki Inoue (The University of Tokyo), Hideaki Kakeya (Kyoto University), Shuji Akai (Osaka University), Jun Terao (The University of Tokyo), Midori Arai (Keio University), Nobuhiro Kihara (Kanagawa University), Mr. Shunta Tokutake (Tokyo University of Agriculture), Mr. Takuya Shiroshita (Tokyo University

of Agriculture), and Mr. Yoshiki Tanaka (Tokyo University of Agriculture) for valuable discussions. We are grateful to Prof. Ken Motokura (Yokohama National University) for his advice on the analysis of solid catalysts. We also thank the Catalysis Society of Japan, Tosoh Co., Ltd., Süd-Chemie Catalysts Japan Inc., JGC Catalysts and Chemicals Ltd., and Fuji Silysia Chemical Ltd. for their generous gifts of porous solid materials, such as zeolite, silica-alumina, alumina, and mesoporous silica. Further thanks to Dr. Emako Suzuki for donating the ketone compounds for steroids **5 s** and **5t**. This work was supported by JSPS KAKENHI Grant Number JP24K17689 (S.S.) and by Whiterock Foundation (M.O.).

## Author contributions

S.S. performed all the experiments described in this paper and was the main initiator of this idea with M.O. S.S. and M.O. prepared the manuscript.

## Competing interests

The authors declare no competing interests.
