## [Transparent Peer Review file · Nature Communications]

Sustainable Purification-Free Synthesis of N–H Ketimines by Solid Acid Catalysis

Corresponding Author: Dr Shintaro Shibata

Version 0:

Reviewer comments:

Reviewer #1

(Remarks to the Author)

Manuscript by Shibata and Onaka reported a new and sustainable method for the synthesis of N-H ketimines. The method is simple and scalable without the need of purification, and especially useful for large scale preparation of different ketimines. The manuscript is well presented, and the contents are suitable for publication in Nature Communication after minor revision.

- 1) Figures 4c and 4d, the reaction from compound 1 to compound 3, please use different R groups (e.g., R, R', R'' etc.) to represent different substituents to discriminate different ketones and amines.
 - 2) Provision of melting points for stable solid samples are highly appreciated.
 - 3) SEM experiments for solid catalysts such as mesoporous alumina containing sulfate ions are highly appreciated.
- Overall, the quality of the experiments is high, and the manuscript is suitable for publication after minor revision.

Reviewer #2

(Remarks to the Author)

Shibata and Onaka described in this manuscript a nice protocol for the synthesis of nitrogen-protected imines, N-H ketimines. In contrast to nitrogen-protected imines, N-H ketimines are more versatile building blocks in chemical synthesis, but are much less unstable and therefore difficult to be synthesized and purified. It is highly desirable to develop an efficient method for the synthesis of N-H ketimines with high purity. Known methods are not versatile and often involve tedious operations. The dehydration-condensation of ketones with ammonia constitutes a very straightforward method for the synthesis of N-H ketimines, but it suffers from high-temperature and high-pressure conditions or strongly acidic dehydrating agents. The authors modified the protocol for the dehydration-condensation of ketones with ammonia by generating in situ a stoichiometric amount of ammonia from hexamethyldisilazane and water and using inorganic solid acid catalysts. The reaction employed readily available starting materials as well as non-toxic and reusable catalysts, underwent under neat conditions, and exhibited broad substrate scope. At the end of the reaction, the solid acid catalyst was separated by filtration and the resulting mixture was subjected to concentration techniques to remove low-boiling-point components, such as the remaining hexamethyldisilazane and byproduct hexamethylsiloxane, delivering a range of N-H ketimines with high purity. The reaction is amenable to large-scale production. Treatment of the N-H ketimine products with hydrogen chloride afforded the corresponding salts quantitatively without producing ammonium chloride as a byproduct. The authors further extended this reaction to the one-pot derivatization of N–H ketimines to N–R ketimines, α -aminonitriles, and hydantoins, and all the products were purified simply by filtration or concentration. Overall, the authors have developed a synthetically useful method for the synthesis of N-H ketimines. The manuscript as well as the Supporting Information were well prepared. In view of the significance of the work in the synthetic methodology, I recommend publication of this work in Nature Communications.

Reviewer #3

(Remarks to the Author)

This paper presents an efficient method for synthesizing nitrogen-protected ketimines (N–H ketimines). While this catalytic system has practical value, the paper reads more like an applied study, primarily reporting the discovery of a catalytic system that facilitates the efficient synthesis of N–H ketimines. However, the authors do not deeply investigate why this catalytic

system works, which is the key question in chemistry. The focus of the manuscript should be on exploring the catalytic mechanism, which is not adequately discussed here. Additionally, the authors do not provide a thorough examination of the stability of the catalytic system or the factors influencing the catalyst. For these reasons, I believe the paper does not meet the standards expected for publication in Nature Communications. I recommend that the authors further explore the mechanism and provide a deeper analysis of the experimental data.

Detailed Comments:

- 1. Lack of Discussion on the Catalytic Mechanism:** The paper presents a novel catalytic system that efficiently synthesizes N–H ketimines, but it does not delve into the mechanism behind the reaction. The authors should investigate how the catalyst influences the reaction mechanism. While the paper mentions solid acid catalysis, it does not clarify whether the solid acid catalysis is truly responsible for the reaction. For instance, basic alumina (Alumina) also efficiently catalyzes the reaction, raising the question of whether the mechanism is really acid-catalyzed. Moreover, the role of sulfur-modified alumina in catalysis is not discussed. The paper should provide more insights into these aspects. Additionally, the data show that both electron-withdrawing and electron-donating groups result in efficient N–H ketimine formation, which does not support the ion-pair mechanism but may indicate a free-radical mechanism instead. The authors should further explore the possibility of a free-radical mechanism.
- 2. Poor Readability of Figure 1:** Figure 1a is overloaded with information and uses excessive colors, making it difficult to discern the differences between previous studies and the current work. The figure should be redesigned to make it more concise and readable. Figure 1b does not clearly illustrate the reaction pathway. It would be helpful to simplify and clearly mark the reaction steps and key intermediates. Figure 1c is unclear and does not effectively convey the intended information. The visualization of the figure should be improved to highlight the main points.
- 3. Catalyst Characterization Issues in Figure 2a:** the catalytic activity of Y-zeolite is presented, but the acidity strength of the zeolite is not characterized. The authors should examine the different acid strengths of the catalysts to explain how the type and strength of acid sites influence the reaction. Moreover, varying the Si/Al ratio affects the pore size of zeolite, which could influence the diffusion of substrates and products in the pores. BET surface area analysis should be performed to further investigate this aspect.
- 4. Catalyst Stability and Reusability:** The paper mentions that alumina with a Si/Al ratio lower than 10 is unstable. The recycled catalysts and the fresh catalysts should be characterized for their Si/Al ratios, and their cycling performance should be examined.
- 5. Role of Water in the Reaction:** The authors mention that water is required for the reaction, but the data show that the amount of water used is far greater than the stoichiometric amount relative to HMDS (water is 15 mol% and HMDS is 120 mol%), suggesting that water may not be a stoichiometric reactant. Water could potentially play a catalytic role in the reaction. The authors should further investigate the influence of water and clarify its role in the reaction.
- 6. Questioning the Acid Catalysis Mechanism:** The paper suggests that the reaction follows a Lewis acid-catalyzed condensation mechanism, but the data in Figure 3a (2a–2q) show that both electron-donating and electron-withdrawing substrates yield high efficiency, which does not support the ion-pair mechanism. This data might suggest a free-radical mechanism instead. The authors should investigate the reaction mechanism in more depth, particularly the potential for a free-radical mechanism.
- 7. Lack of Research on the Effect of Sulfate Modification:** In Figure 2c, the authors do not study how sulfate modification promotes the catalytic activity. How does sulfate modification enhance the catalyst's activity? This should be addressed in the reaction pathway analysis.
- 8. Insufficient DFT Modeling:** The authors mention using DFT modeling to study the catalytic mechanism, but the results do not provide clear insights into how the catalyst affects the reaction. The DFT model should be more detailed and show how the catalyst interacts with the substrate and the changes in energy along the reaction pathway to provide deeper mechanistic insights.

Version 1:

Reviewer comments:

Reviewer #1

(Remarks to the Author)

The authors have made necessary revisions, and the manuscript is suitable for publication from my view point.

Reviewer #3

(Remarks to the Author)

The authors have thoroughly addressed all the concerns I previously raised. I recommend the revised manuscript for publication in Nature Communications.

Response to Reviewers

We sincerely thank the reviewers for their valuable and constructive comments on our manuscript. Below, we provide point-by-point responses to each of their remarks. All revisions in the main text and the Supplementary Information are highlighted in the revised versions for ease of reference, and a concise summary of these revisions is presented after the responses.

Reviewer comments

Reviewer #1 (Remarks to the Author):

Manuscript by Shibata and Onaka reported a new and sustainable method for the synthesis of N-H ketimines. The method is simple and scalable without the need of purification, and especially useful for large scale preparation of different ketimines. The manuscript is well presented, and the contents are suitable for publication in Nature Communication after minor revision.

- 1) Figures 4c and 4d, the reaction from compound 1 to compound 3, please use different R groups (e.g., R, R', R'' etc.) to represent different substituents to discriminate different ketones and amines.
- 2) Provision of melting points for stable solid samples are highly appreciated.
- 3) SEM experiments for solid catalysts such as mesoporous alumina containing sulfate ions are highly appreciated.

Overall, the quality of the experiments is high, and the manuscript is suitable for publication after minor revision.

Response #1

We thank the reviewer for their insightful comments. Below, we provide responses one by one:

- 1) We have rechecked Fig. 1–5 and revised the structures as suggested. Different R groups (e.g., R, R', R'') are now used to clearly distinguish the various ketones and amines.
- 2) We have added the following explanation at the beginning of Section 9.1 in the Supplementary Information regarding the melting points of the solid samples: "The solid samples of N–H ketimines and their hydrochloride salts are thermally highly unstable. Even when gently heated under conventional Schlenk conditions, complete prevention of hydrolysis to the corresponding ketones is difficult. Decomposition often occurs before melting, and discoloration is frequently observed. Therefore, accurate melting point measurements were not be able to be performed. It is considered that extremely strict anhydrous and oxygen-free conditions would be required for such measurements."

In other words, it is difficult to determine the temperature at which thermal decomposition begins, and the composition of the samples after heating clearly differs from that before heating, making accurate melting point measurements impossible.

3) In addition to the suggested SEM analysis, we performed IR-ATR, XRD, XPS, and EDS measurements, providing a more detailed characterization than previous studies. The results are included in Section 2.3 of the Supplementary Information.

Reviewer #2 (Remarks to the Author):

Shibata and Onaka described in this manuscript a nice protocol for the synthesis of nitrogen-protected imines, N-H ketimines. In contrast to nitrogen-protected imines, N-H ketimines are more versatile building blocks in chemical synthesis, but are much less unstable and therefore difficult to be synthesized and purified. It is highly desirable to develop an efficient method for the synthesis of N-H ketimines with high purity. Known methods are not versatile and often involve tedious operations. The dehydration-condensation of ketones with ammonia constitutes a very straightforward method for the synthesis of N-H ketimines, but it suffers from high-temperature and high-pressure conditions or strongly acidic dehydrating agents. The authors modified the protocol for the dehydration-condensation of ketones with ammonia by generating in situ a stoichiometric amount of ammonia from hexamethyldisilazane and water and using inorganic solid acid catalysts. The reaction employed readily available starting materials as well as non-toxic and reusable catalysts, underwent under neat conditions, and exhibited broad substrate scope. At the end of the reaction, the solid acid catalyst was separated by filtration and the resulting mixture was subjected to concentration techniques to remove low-boiling-point components, such as the remaining hexamethyldisilazane and byproduct hexamethylsiloxane, delivering a range of N-H ketimines with high purity. The reaction is amenable to large-scale production. Treatment of the N-H ketimine products with hydrogen chloride afforded the corresponding salts quantitatively without producing ammonium chloride as a byproduct. The authors further extended this reaction to the one-pot derivatization of N-H ketimines to N-R ketimines, α -aminonitriles, and hydantoins, and all the products were purified simply by filtration or concentration. Overall, the authors have developed a synthetically useful method for the synthesis of N-H ketimines. The manuscript as well as the Supporting Information were well prepared. In view of the significance of the work in the synthetic methodology, I recommend publication of this work in Nature Communications.

Response #2

We sincerely thank the reviewer for carefully reading our manuscript. We are also very grateful for the positive evaluation of the significance of our work, the methodology, and the quality of the

Supporting Information.

Reviewer #3 (Remarks to the Author):

This paper presents an efficient method for synthesizing nitrogen-unprotected ketimines (N–H ketimines). While this catalytic system has practical value, the paper reads more like an applied study, primarily reporting the discovery of a catalytic system that facilitates the efficient synthesis of N–H ketimines. However, the authors do not deeply investigate why this catalytic system works, which is the key question in chemistry. The focus of the manuscript should be on exploring the catalytic mechanism, which is not adequately discussed here. Additionally, the authors do not provide a thorough examination of the stability of the catalytic system or the factors influencing the catalyst. For these reasons, I believe the paper does not meet the standards expected for publication in Nature Communications. I recommend that the authors further explore the mechanism and provide a deeper analysis of the experimental data.

Response #3

Thank you for your careful review of our manuscript and for your worthwhile comments. Our study is based on the classic dehydration condensation of a ketone and ammonia, a reaction that effectively utilizes the function of a solid acid catalyst. However, as you rightly pointed out, the original manuscript lacked sufficient material analysis of the catalyst and did not adequately explain its structure and catalytic activity.

In response, we have performed a detailed material analysis of the catalyst. Our catalyst is similar to conventional catalysts referenced during its preparation, and we have concluded that the introduction of sulfate groups to the amorphous and porous alumina catalyst improves its activity, as has been previously suggested.

We have incorporated these changes and additional data into our responses below. We hope you find our revised manuscript to your satisfaction.

Detailed Comments:

1. Lack of Discussion on the Catalytic Mechanism: The paper presents a novel catalytic system that efficiently synthesizes N–H ketimines, but it does not delve into the mechanism behind the reaction. The authors should investigate how the catalyst influences the reaction mechanism. While the paper mentions solid acid catalysis, it does not clarify whether the solid acid catalysis is truly responsible for the reaction. For instance, basic alumina (Alumina) also efficiently catalyzes the reaction, raising the question of whether the mechanism is really acid-catalyzed. Moreover, the role of sulfur-modified alumina in catalysis is not discussed. The paper should provide more insights into these aspects.

Additionally, the data show that both electron-withdrawing and electron-donating groups result in efficient N–H ketimine formation, which does not support the ion-pair mechanism but may indicate a free-radical mechanism instead. The authors should further explore the possibility of a free-radical mechanism.

Response (1)

The condensation reaction between ketones and ammonia, which is classically considered as acid-catalyzed ionic reaction. Our catalyst screening results shown in Fig. 2a further support this conclusion: when we compared different catalysts with well-defined acid sites, polymer-supported acids exhibited only minimal conversion, while amines induced no reaction at all. Similar trends are observed for the solid acids with the same framework, where the acidic H-Y zeolite is significantly more active than the basic Na-Y bearing exchanged alkali-metal ions. These results infer that, even in the alumina cases, the reaction proceeds on the acidic sites of the alumina surface.

According to the surface analyses of alumina reported in Refs. 35–37, potential active sites of alumina highly active three-coordinated Lewis acid sites, or Brønsted acid sites formed by bridging hydroxyl groups near these Lewis acid centers. However, in the base-addition experiment shown in Fig. 2c, when the proton scavenger of proton sponge was added, the reaction was hardly affected, concluding that the true active sites for this reaction are Lewis acid centers.

The effect of sulfate ions in alumina is evident from the comparison between the catalysis of *meso*-Al₂O₃ and *meso*-Al₂O₃/SO₄²⁻ prepared by the same method (Fig. 2a), with the sulfate-containing catalyst showing higher activity. Furthermore, combining newly conducted IR-ATR, XRD, XPS, SEM, and EDS analyses in addition to the previous characterization results indicates that most of sulfate ions are uniformly dispersed within the alumina framework, and their incorporation near Lewis acid sites enhances the activity of these centers.

Regarding the substrate scope (Fig. 3c), reaction conditions were optimized for each substrate to allow quantitative isolation of N–H ketimines. Reactions under solvent-free conditions proceeded significantly faster than those in solvent. Comparison of reactions in solvent revealed that substrates bearing electron-donating groups (**2i**, **2p**) required higher catalyst loadings and elevated temperatures than those bearing electron-withdrawing groups (**2m**, **2o**). For substrate **2q**, the reaction temperature was slightly increased to improve the solubility of **2q**, as the product precipitated during the reaction.

Although not reported in the main text, adding 20 mol% of a radical scavenger, TEMPO (2,2,6,6-tetramethylpiperidin-1-yl)oxyl, under the conditions of Fig. 2a did not reduce the product yield. This further confirms that the reaction proceeds via an acid–base ionic mechanism rather than a radical pathway.

Taken together, these results indicate that the reaction is catalyzed via an ionic mechanism on the Lewis acid sites of alumina.

2. Poor Readability of Figure 1: Figure 1a is overloaded with information and uses excessive colors, making it difficult to discern the differences between previous studies and the current work. The figure should be redesigned to make it more concise and readable. Figure 1b does not clearly illustrate the reaction pathway. It would be helpful to simplify and clearly mark the reaction steps and key intermediates. Figure 1c is unclear and does not effectively convey the intended information. The visualization of the figure should be improved to highlight the main points.

Response (2)

In response to the reviewer's comments, Fig. 1a–c have been revised for improved readability, making the catalytic cycle and key points of the reaction easier to understand. The color scheme has been slightly adjusted, and the layout has been optimized for clarity.

3. Catalyst Characterization Issues in Figure 2a: the catalytic activity of Y-zeolite is presented, but the acidity strength of the zeolite is not characterized. The authors should examine the different acid strengths of the catalysts to explain how the type and strength of acid sites influence the reaction. Moreover, varying the Si/Al ratio affects the pore size of zeolite, which could influence the diffusion of substrates and products in the pores. BET surface area analysis should be performed to further investigate this aspect.

Response (3)

Although zeolites were not the optimal catalysts in this study and thus received little attention, as the reviewer pointed out, we compared the acid strength of proton-type zeolites with different pore structures. When the amount of acid (i.e., Si/Al ratio) was similar, zeolites with lower acidity tended to exhibit higher catalytic activity. We have cited a reference comparing acid strengths based on zeolite structures (Ref. 31) and added this discussion to the main text. While no clear trend was observed dependent on differences in surface area, the BET surface areas of the proton-type zeolites and other solid acid catalysts used were added to Section 1 of the Supplementary Information.

4. Catalyst Stability and Reusability: The paper mentions that alumina with a Si/Al ratio lower than 10 is unstable. The recycled catalysts and the fresh catalysts should be characterized for their Si/Al ratios, and their cycling performance should be examined.

Response (4)

Alumina does not contain silicon, so the Si/Al ratio is not applicable. Furthermore, the manuscript does not mention that “Si/Al ratio lower than 10 is unstable”. Regarding the recycling of commercially available γ -alumina catalysts, the experiments shown in Fig. 4a were conducted on a relatively large laboratory scale (100 g). After the reaction, the used catalyst was calcined at 600 °C and reused under the same reaction conditions, and no loss of catalytic activity was observed, as also shown in the result of **2a** in the yield of >99% in parentheses. γ -Alumina is thermally stable at approximately 600 °C, and its simple regeneration by calcination in the air represents a significant advantage for large-scale synthesis.

5. Role of Water in the Reaction: The authors mention that water is required for the reaction, but the data show that the amount of water used is far greater than the stoichiometric amount relative to HMDS (water is 15 mol% and HMDS is 120 mol%), suggesting that water may not be a stoichiometric reactant. Water could potentially play a catalytic role in the reaction. The authors should further investigate the influence of water and clarify its role in the reaction.

Response (5)

As shown in Fig. 2b, the reaction does not proceed without the addition of water, demonstrating that water plays a crucial role in hydrolyzing HMDS to generate ammonia species. The generated ammonia then condenses with the ketone to yield an *N*-H ketimine and water. This newly formed water reacts with HMDS to produce siloxane and additional ammonia, thereby sustaining the catalytic cycle depicted in Fig. 1b and driving the reaction forward. As the reaction progresses, up to 100 mol% of water is generated from the ketone. Therefore, even when 15 mol% of water is initially added, as shown in Fig. 2b, the total water in the system (115 mol%) never exceeds the amount of HMDS (120 mol%). These results clearly indicate that water is not required in a stoichiometric amount, but rather in a catalytic quantity. Thus, we conclude that water functions catalytically to initiate the reaction. Furthermore, within the substrate scope shown in Fig. 3, for less reactive ketones, the amounts of not only the catalyst but also water and HMDS are appropriately adjusted to achieve efficient conversion.

6. Questioning the Acid Catalysis Mechanism: The paper suggests that the reaction follows a Lewis acid-catalyzed condensation mechanism, but the data in Figure 3a (2a–2q) show that both electron-donating and electron-withdrawing substrates yield high efficiency, which does not support the ion-pair mechanism. This data might suggest a free-radical mechanism instead. The authors should investigate the reaction mechanism in more depth, particularly the potential for a free-radical mechanism.

Response (6)

As mentioned in Response 1, the addition of 20 mol% of the radical scavenger TEMPO (2,2,6,6-tetramethylpiperidin-1-yl)oxyl) under the conditions of Fig. 2a had no effect on the product yield, indicating that a radical pathway is not involved. Regarding the substituent effects (Fig. 3c), under comparable solvent-based conditions, substrates bearing electron-donating groups (**2i**, **2p**) required higher catalyst loadings and elevated temperatures than those bearing electron-withdrawing groups (**2m**, **2o**). For substrate **2q**, the reaction temperature was slightly increased to prevent precipitation of the poorly soluble product; however, the reaction proceeded nearly quantitatively even at 60 °C.

Overall, these results indicate that the reaction is catalyzed via an acid–base ionic mechanism, with no involvement of a radical pathway.

7. Lack of Research on the Effect of Sulfate Modification: In Figure 2c, the authors do not study how sulfate modification promotes the catalytic activity. How does sulfate modification enhance the catalyst's activity? This should be addressed in the reaction pathway analysis.

Response (7)

Based on the additional material analyses conducted after the review, as well as the results from similar catalysts reported in previous studies (Ref. 32), we consider that the incorporation of sulfate into the alumina framework enhances the catalytic activity of Lewis acid sites (see also our response to Comment 1). In Ref. 32, it is suggested that sulfate is incorporated into part of the alumina structure as O–Al–O–SO₂–O–Al–O, which increases the acid strength of neighboring Al centers. We believe that the catalyst prepared in this study adopts a similar structure. In addition to this, IR-ATR, XRD, SEM, and XPS data obtained after receiving the reviewers' comments, it appears that the sulfate is not simply a physical mixture of Al₂(SO₄)₃ and alumina, but that the sulfate moiety is rather uniformly dispersed throughout the catalyst. These results further support the incorporation of sulfate into the alumina framework.

On the other hand, the reaction proceeds well even with γ -Al₂O₃ or mesoporous alumina that do not

contain sulfate ions, indicating experimentally that the sulfate moiety primarily serves to enhance the surface acidity of alumina. In Fig. 2c, the reaction pathway analysis was intended to investigate whether NH_3 or TMSNH_2 , generated from the decomposition of HMDS and water, is the reactive species. To fully examine the effect of sulfate, larger-scale calculations would be required, which are currently beyond our study scope.

8. Insufficient DFT Modeling: The authors mention using DFT modeling to study the catalytic mechanism, but the results do not provide clear insights into how the catalyst affects the reaction. The DFT model should be more detailed and show how the catalyst interacts with the substrate and the changes in energy along the reaction pathway to provide deeper mechanistic insights.

Response (8)

DFT calculations on the catalytic reaction over an alumina cluster, constructed with reference to the surface structures reported in Refs. 35–37, are presented in Section 8 of the Supplementary Information. The primary purpose of these calculations is to clarify whether the reactive species is NH_3 or TMSNH_2 . Comparison with uncatalyzed conditions indicates that the presence of the catalyst generally lowers the activation energy (Fig. S11). The energy diagram in Fig. S11 shows that the pathway from ketone and ammonia to *N*-H ketimine and water proceeds overall in an energetically favorable (downhill) manner. Furthermore, the water formed reacts with HMDS to generate highly stable siloxane and ammonia, further lowering the energy of the product state and enabling smooth progression of the catalytic cycle (see Fig. S10 for single-molecule calculations confirming the stability of siloxane formation). It should be noted that a more detailed DFT analysis that faithfully reproduces the entire catalytic cycle would require very large-scale calculations, which is not practically feasible within the scope of the present study.

Summary of Revisions in the Main Manuscript

We have re-checked the submission format and adjusted it appropriately, and we have revised the manuscript based on the reviewers' comments. Below, we provide a concise summary of the main revisions made in the manuscript (see details above in the point-by-point responses).

- **p.3:** Revised Fig. 1a–c for clarity, with Fig. 1b changed to a catalytic cycle diagram and the caption updated with explanations of chemical terms.
- **p.4, l.76:** Added a description of the methanolysis of HMDS.
- **p.4, l.83–90:** Added a description of the difference in reactivity arising from the structural variations of zeolites.
- **p.7, l.140–145:** Added a description of the catalyst analysis.
- **p.7, l.165–168:** Added results of HMDS methanolysis experiments in Section 3.4 of the Supplementary Information and revised the text accordingly.
- **pp. 8, 10, and 12; Figs. 3, 4, and 5:** Revised the schemes, changing R to R' or R''.
- **p.13, 297–301:** Added one sentence to the data availability.
- **p.15, l.371–372:** Added Ref. 31, which shows the acid strength sequence depending on differences in zeolite pore structures, and renumbered the in-text citations accordingly.
- **p.15, l.417–418:** Added one sentence to the acknowledgments.

Summary of Revisions in the Supplementary Information

We also summarize the major revisions made in the Supplementary Information. We incorporated the results of additional experiments, but no revisions were made that alter the essence of this study. We also corrected minor errors that were newly identified; however, these are limited in nature and do not affect the chemical interpretations or the conclusions.

- **p.2:** Added Sections 2.3 and 3.3 to the Table of Contents.
- **p.3:** Added the instruments used for additional analyses and the specific surface area of the solid materials.
- **p.11:** Changed the section title and added a description of the preparation method of *meso*-Al₂O₃ in the last sentence.
- **p.12:** Corrected the calcination time in the caption of Fig. S1.
- **pp. 12–14:** Added the analysis results of *meso*-Al₂O₃ and the pore size distribution plots of each catalyst (Fig. 3).
- **pp. 15–23:** Added Section 2.3 and included the analysis results of *meso*-Al₂O₃/SO₄²⁻ obtained by IR-ATR, XRD, XPS, and SEM-EDS measurements.

- **p.26:** Added additional experiments in Section 3.3 showing the rate of HMDS methanolysis under catalytic conditions.
- **p.27:** Revised part of the description in the Experimental section.
- **p.28:** Added a detailed Experimental section on the reuse experiment with re-calcination at the 100 g scale of ketone 1a.
- **pp. 29–30:** Corrected the adsorbed water content of neutral silica from 0.4 mmol/g to 4 mmol/g, and revised the drying temperature in the footnote of Table S3 to 120 °C.
- **p.44:** Added an explanation at the beginning on why measuring the melting points of *N*-H ketimine compounds is difficult.
- **p.87:** Corrected an error in the calculated exact mass.
- **p.98:** Corrected the compound number in the title from **2a** to **2a·HCl**.
- **p.291:** Corrected an exact mass-calculation error in the figure (same as on p.87).
- **p.410:** Added Ref. 15 showing the XPS database, and renumbered the in-text citations accordingly.